# Modification of Multiwalled Carbon Nanotubes and Their Mechanism of Demanganization

**DOI:** 10.3390/molecules28041870

**Published:** 2023-02-16

**Authors:** Yuan Zhou, Yingying He, Ruixue Wang, Yongwei Mao, Jun Bai, Yan Dou

**Affiliations:** 1School of Water and Environment, Chang’an University, No. 126 Yanta Road, Xi’an 710054, China; 2Key Laboratory of Subsurface Hydrology and Ecological Effects in Arid Region of the Ministry of Education, Chang’an University, No. 126 Yanta Road, Xi’an 710054, China

**Keywords:** M-MWCNTs, Mn(II) removal, kinetic model, isotherm model, PSO-BP model

## Abstract

Multiwalled carbon nanotubes (MWCNTs) were modified by oxidation and acidification with concentrated HNO_3_ and H_2_SO_4_, and the modified multiwalled carbon nanotubes (M-MWCNTs) and raw MWCNTs were characterized by several analytical techniques. Then the demanganization effects of MWCNTs and M-MWCNTs were well investigated and elucidated. The experimental data demonstrated that the adsorption efficiency of Mn(II) could be greatly promoted by M-MWCNTs from about 20% to 75%, and the optimal adsorption time was 6 h and the optimal pH was 6. The results of the kinetic model studies showed that Mn(II) removal by M-MWCNTs followed the pseudo-second-order model. Isothermal studies were conducted and the results demonstrated that the experimental data fitted well with the three models. The reliability of the experimental results was well verified by PSO–BP simulation, and the present conclusion could be used as a condition for further simulation. The research results provide a potential technology for promoting the removal of manganese from wastewater; at the same time, the application of various mathematical models also provides more scientific ideas for the research of the mechanism of adsorption of heavy metals by nanomaterials.

## 1. Introduction

Manganese, a heavy metal, is abundant in nature and also plays a significant role in many important industries [1,2]. However, due to factors such as artificial mining of mineral resources, illegal discharge of pollutants from factories, and dissolution of manganese minerals in aquifers caused by changes in ecological environment, the problem of pollution has become increasingly prominent [3,4,5]. In recent years, the concentration of manganese in underground wells in many countries has far exceeded the standards of the World Health Organization [6,7,8,9]. Overexposure to manganese can cause a variety of negative health effects for humans [10,11], and high manganese content in plants will lead to crop necrosis and cotton wrinkling, thus affecting the food and textile industry [12]. Studies have shown that in manganese-polluted areas, the soil and groundwater are usually containing acidic organic matter, so manganese mostly exists in the form of divalent ions [13,14,15].

There have been some studies on methods for the removal of manganese from a solution, such as the oxidation precipitation method [16], the ion exchange method [17], the reverse osmosis method [18] and the adsorption method, which is the more commonly used removal method [19,20,21]. Recently, with the development of nanomaterials, carbon nanotubes have been reported as the new adsorbents for the removal of heavy metal and organic pollution, such as chlorobenzenes, herbicides, heavy metal ions (Pb^2+^ and Ca^2+^), and inorganic nonmetallic ions, including F^−^ [22,23].

Due to lower synthesis and purification costs, and easy application to water treatment, multi-walled carbon nanotubes (MWCNTs) are more widely used than single-walled carbon nanotubes (SWCNTs) [24]. To improve the removal of heavy metals by carbon nanotubes, the original MWCNTs usually need to be modified. The ordinary modification method is to use a strong oxidant to oxidize the MWCNTs under reflux or ultrasonic conditions [25,26]. This oxidation endows carbon nanotubes with rich, oxygen-containing groups and exposes the adsorption site [17,27]. Meanwhile, in order to solve complex nonlinear problems in practice, some neural network models, such as artificial neural networks (ANN) and backpropagation-based training optimization neural networks (BPNN), have been gradually applied in the field of pollutant removal [28,29,30].

It has been reported that MWCNTs can adsorb manganese [31], but the removal efficiency, adsorption mechanism, and related experimental verification work have not been clearly carried out. Therefore, the objectives of this work are as follows: (1) to modify raw MWCNTs and to characterize the MWCNTs and M-MWCNTs by SEM, XPS, FT-IR, etc.; (2) to study the effects of demanganization on MWCNTs and M-MWCNTs with varying pH, contact time, and temperature; (3) to describe the characteristics of the Mn(II) removal by the MWCNTs and M-MWCNTs with an adsorption kinetic model and an isotherm model; and (4) to simulate the adsorption process by using PSO-BP modeling.

## 2. Results and Discussion

### 2.1. Characterization of MWCNTs and M-MWCNTs

#### 2.1.1. FT-IR

The most important use of infrared spectroscopy (FTIR) is the structural analysis of organic compounds [32]. In this research, FT-IR was used to verify the structural analysis of MWCNTs and M-MWCNTs, and M-MWCNTs were modified by oxidation and acidification. The FTIR spectra of Figure 1 illustrated that there were functional groups, −OH groups (3200~3600 cm^−1^), −C=O− groups (1600 cm^−1^), and −C−C− groups (1150 cm^−1^), on the external and internal surface of MWCNTs and M-MWCNTs [33]. The transmittance (%) of the −OH groups and −C=O− groups in M-MWCNTs were stronger than in raw MWCNTs; the modification increased the active sites on the surface and further altered the surface polarity and charges. It is reported that the stretching vibration absorption peak of −C=O− usually appears in 1755–1670 cm^−1^. In this study, an absorption peak of −C=O− is observed at 1623 cm^−1^. This phenomenon has also shown up in the work of other researchers [34], possibly because the conjugate effect of carbon nanotubes makes the absorption of −C=O move to the shortwave direction [35].

#### 2.1.2. XPS

XPS can be used to analyze elements present on the surface of the sample and bonding species [36]. The XPS wide-scan spectrum (Figure 2a) shows that the elements present on the surface of MWCNTs and M-MWCNTs were mainly C and O. New active sites are provided, as evidenced by the increased oxygen content of M-MWCNTs. This indicates that the adsorption capacity of M-MWCNTs for heavy metals will be enhanced [37]. The increase in O content (from 3.5% to 13.66%) showed that the modification was successful. Figure 2b shows the fitted XPS spectra of the O1s of M-MWCNTs compared to the reference XPS of MWCNTs and M-MWCNTs; the experimental data showed that −C=O− groups (530.7 eV) and −C−O− groups (533.6 eV) were on the surface of the M-MWCNTs.

#### 2.1.3. SEM

The morphology investigation of MWCNTs and M-MWCNTs was performed using SEM (Figure 3). Figure 3b shows that the MWCNTs and M-MWCNTs were about 20 nm in diameter, M-MWCNTs were shorter than MWCNTs, and reunions were more likely to occur in M-MWCNTs particles, a phenomenon that was consistent with the Zeta potential results.

#### 2.1.4. Zeta Potentials

The zeta potentials were measured as a function of pH to determine the pH_PZC_ of MWCNTs and M-MWCNTs [38]. The results (Figure 4) showed that the zeta potentials of MWCNTs and M-MWCNTs decreased with the increasing pH, whereas the zeta potentials of M-MWCNTs became more negative after the treatment. The zeta potentials of M-MWCNTs were all less than 0. It can be speculated that the reason for this phenomenon was the influence of certain groups (−COOH, −OH) [31]. Surface negativity is favorable for the adsorption of heavy metal ions from the solution by the adsorbent [39].

#### 2.1.5. Size

Figure 5 shows the size distribution of MWCNTs and M-MWCNTs at pH 6.2. In general, the dynamic light-scattering results from MWCNTs represent agglomerations rather than individual nanomaterials [40]. In the measurement process, the agglomeration of the original MWCNTs leads to a wider particle size distribution and an increase in the mean value (Figure 5a). This phenomenon becomes more obvious over time (Figure 5a blue line). The average particle size of M-MWCNTs treated with mixed acid decreased, and the result of the three measurements was close to 193.5, indicating that the stability of carbon nanotubes in an aqueous solution was significantly improved. It is remarkable that, compared with the raw M-MWCNTs, the particle size of the M-MWCNTs with adsorbed Mn(II) was increased significantly; the average hydrated particle size increased from 193.5 nm to 320 nm.

### 2.2. Effect of Contact Time

The effect of contact time was studied under the following conditions: agitation speed, 180 rpm; adsorbent, 20 mg; initial concentration, 5 mg/L; pH, 5.6; temperature, 25 °C; and the mass ratio of adsorbent in solution was 1 g/L. The results are presented in Figure 6. It can be observed that the adsorption of Mn(II) onto the MWCNTs reached equilibrium rapidly within 1 h, but the removal efficiency of Mn(II) was low (27.9%). The adsorption of Mn(II) by M-MWCNTs increased rapidly in the first 30 min, then increased at a slower rate and reached equilibrium at 6 h. This result is corroborated in the literature [31].

### 2.3. Effect of pH

The effect of pH on the demanganization by MWCNTs and M-MWCNTs is shown in Figure 7. The pH value varied from 2 to 9, contact time was 10 h, mass of adsorbent was 20 mg, concentration of Mn(II) was 5 mg/L, agitation speed was 180 rpm, the mass ratio of adsorbent in solution was 1 g/L, and the experimental temperature was 25 °C. The results indicated that the removal efficiency of Mn(II) increased. When the pH value is higher than 9, Mn(II) should be formed and Mn(OH)_2_ can even be precipitated [41]. In addition, under acidic conditions, H^+^ ions compete for the active sites with Mn(II) ions. As the pH increases, the concentration of H^+^ ions decreases, leaving more adsorption sites for Mn(II) ions. As the pH continues to increase, the Mn(II) is hydrolyzed, forming Mn(OH)^+^, Mn(OH)_2_, Mn_2_(OH)^+^_3_, and Mn(OH)^−^_4_. When pH > 8, the Mn(II) begins to form a precipitate [42].

When pH < 3, the adsorption rate of Mn(II) onto raw MWCNTs is close to 0. In the strong acid solution, the adsorption effect of raw MWCNTs on Mn(II) was inhibited, mainly due to a competitive effect between H^+^ and Mn^2+^ at the active site. The zeta potential results also support this conclusion; experimental zeta potential results showed that the zeta potential of the raw MWCNTs was positive in a solution of pH < 3 [43].

### 2.4. Kinetic Modeling

The results of the three kinetic models are shown in Figure 8a,b, and the parameters are shown in Table 1. The results revealed that the adsorption of Mn(II) onto MWCNTs and M-MWCNTs followed second-order kinetics, which suggests that the adsorption process before both is chemical adsorption. The fitting line of the Weber–Morris model (Figure 8c) showed that the adsorption of Mn(II) onto MWCNTs and M-MWCNTs could be described in two or three stages, indicating that the Mn(II) adsorption process is controlled by some diffusion mechanisms, probably including an intra-particle diffusion mechanism, bulk diffusion, and film diffusion mechanisms.

### 2.5. Isotherm Modeling

The results of experimental data fitting to three adsorption isotherm models for Mn(II) are presented in Table 2 and Figure 9.

The adsorption of Mn(II) by MWCNTs simultaneously conformed to three models; all the values of *R*^2^ were >0.97 in the three isotherms. This indicated that the adsorption of Mn(II) by MWCNTs was complex and contained a variety of mechanisms under experimental conditions; the adsorption process had both monolayer and inhomogeneous surface chemisorption. As for M-MWCNTs, the *R*^2^ of the Langmuir model was close to 1, which is better than others. This indicated that the adsorption process of Mn(II) by M-MWCNTs was monolayer adsorption. As the Mn(II) ions attached to the site of M-MWCNTs (−COOH, −OH), no further adsorption occurred at that site [44]. Consequently, the adsorption capacity reached the maximum when the monolayer adsorption of Mn(II) ions was completely formed on the surface. The value of theoretical adsorption capacity calculated for M-MWCNTs is approximately 10 times higher than that of MWCNTs. The values of *R_L_* were from 0 to 1 for the studied concentration range, which means the adsorption of Mn(II) by M-MWCNTs is considered favorable.

Notably, the MWCNT and M-MWCNT results followed the D–R model well (*R*^2^ = 0.989 and 0.913, respectively). The value of *E_S_* indicated that the MWCNT adsorption process was mainly ion-exchange adsorption, while that of M-MWCNTs was mainly chemical adsorption [45].

### 2.6. PSO–BP Modeling

In this study, 35 and 39 sets of experiments were designed for MWCNTs and M-MWCNTs, respectively, and each set of experiments was conducted thrice. All data (35 × 3, 39 × 3, Table A1) were used for model fitting in order to reveal the inherent mechanisms in the process [46]. Imported data were normalized, randomly shuffled, and divided into three groups for crossover verification (70% for training, 15% testing, and 15% validation).

In Figure 10, two models were developed for adsorption of MWCNTs (Model A) and M-MWCNTs (Model B), to choose the optimal network structure, 1–14 neurons were applied in the hidden layer, RMSE and R2 were used to evaluate the effectiveness of PSO-BP, and in the selected range, Figure 10 shows that the calculated results of the model were consistent with the experimental data, and for the value of R2, 11 of the 14 neurons had values around 0.95 in Figure 10a, and 12 of the 14 neurons of values ranged from 0.95 to 1.00 in Figure 10b. For the value of RMSE, in Figure 10a, 11 values were between 2.5–3, and in Figure 10b, the 10 values ranged from 3–5. The parameters of the PSO–BP model are listed in Table 3. With this fixed parameter, the optimal weights and biases obtained from the two PSO–BP models are shown in Table A2. Figure A1 shows the comparison between the input and output data of two models. The values of R for the training, testing, validation, and all data of both models were better than 0.97, which demonstrated that the predicted data agreed well with the experimental data using the PSO–BP model. Most of the data were distributed on the line of Y = T, indicating good compatibility of the experimental data with the PSO–BP-forecasted data. To further validate the performance of the model predictions, the four non-linear statistics of the model were evaluated. The results of the evaluation (Table 4) demonstrated that the predictions of both models were statistically significant.

## 3. Materials and Methods

### 3.1. Materials

The raw MWCNTs were purchased from COCC (Chengdu Institute of Organic Chemistry, Chinese Academy of Science). Using carbon gas as the carbon source, this material was produced via the CVD (catalytic vapor decomposition) method; its outer diameter was between 5 and 15 nm, the surface area was in the range of 220–300 m^2^/g, and the purity of the material was above 95%, with low metal impurity content and high electrical conductivity.

Mn(II) stock solution (1000 mg/L) was prepared by dissolving 3.6010 g MnCl_2_•4H_2_O (GR) in 1000 mL of deionized water. All the manganese-containing solutions in later experiments were diluted from the stock solution. The concentration of Mn(II) was determined by atomic absorption spectrometry (WFX120).

### 3.2. Preparation of M-MWCNTs

A total of 5 g of raw MWCNTs was added to a 120 mL mixture of HNO_3_ (68%) and H_2_SO_4_ (98%) (*v*/*v* 1:3). The mixture was sonicated at 40 °C for 4 h and stirred at room temperature for 24 h, so that the MWCNTs were fully in contact with the oxidant and were oxidized. The resulting MWCNTs were separated from the solution using a 0.22 μm membrane filter, rinsed thoroughly with deionized water to remove excess acid, and then rinsed with anhydrous ethanol to accelerate drying. The obtained sample was dried at 65 °C for 24 h. Figure 11 shows the preparation of M-MWCNTs.

### 3.3. Characterization Analysis Methods

The FT-IR spectra of the dried samples were determined by an IRTracer 100 FTIR spectrometer (Shimadzu, Japan), using the potassium bromide powder method. The range of scanning was 4000–350 cm^−1^ and the resolution was set to 4 cm^−1^.

XPS (X-ray photoelectron spectroscopy) was performed to investigate the creation of ionic networks between MWCNTs and M-MWCNTs on ESCALAB Xi+. The main parameters were spatial resolution (1 μm) and energy resolution (0.48 eV). The sample needed to be pressed before testing.

SEM (scanning electron microscopy, Zeiss Ultra Plus) was used to detect the surface morphology of the MWCNTs and M-MWCNTs [47] and the micro area constitution of materials was analyzed using Oxford X-Max 50 mm Energy Disperse Spectroscopy (EDS).

A Malvern Zetasizer Ultra was used to measure the particle sizes and zeta potentials of MWCNTs and M-MWCNTs. The samples were dispersed by ultrasonication for 20 min before measurement.

### 3.4. Batch Experiments

Adsorption experiments. A total of 20 mg MWCNTs and 20 mg M-MWCNTs were separately mixed with 20 mL of Mn(II) solution in a polypropylene centrifuge tube of 50 mL. The initial Mn(II) concentration was 5, 10, 20, 50, 80, and 100 mg/L, while the initial pH value was from 5.6 to 5.2. The mixture was shaken for 10 h and 25 °C at 180 rpm, then filtered using a 0.22 μm membrane filter.

Adsorption thermodynamics experiments. The temperature was set at 25, 35, 45, and 55 °C, and the contact time for adsorption kinetics was controlled from 5 min to 24 h. To study the effect of pH, the pH value was adjusted using sodium hydroxide solution (0.01~0.1 mol/L) and hydrochloric acid solution (0.01~0.1 mol/L).

### 3.5. Theory

#### 3.5.1. Kinetic Modeling

The pseudo-first-order model, pseudo-second-order model, and Weber–Morris model were the kinetic models used to describe the mechanism of Mn(II) adsorption onto MWCNTs and M-MWCNTs. The pseudo-first-order kinetic model can be used to fit the materials with fewer active sites, higher initial adsorption concentration and the adsorption process in the early stage, while the pseudo-second-order kinetic model is just the opposite [48]. Pseudo-first-order and pseudo-second-order dynamics models belong to empirical models. They are simple to calculate, but they fail to explain the adsorption process in terms of mechanism. A Weber–Morris dynamics model has been put to use in the adsorption of metal ions by many scholars [49]. It can be used to analyze the rate-controlling steps of the adsorption reaction, which can guide the adsorption process.

The pseudo-first-order model, which is widely used in kinetic adsorption simulation, is based on the assumption that adsorption is controlled by diffusion steps [50]. Equation (1) describes this model, where $q_{e}$ is the equilibrium absorption capacity, $q_{t}$ is the absorption capacity at time *t*, and $k_{f}$ is the rate constant. Equation (2) is the integrated form.
(1)$$\left(dq_{t})/dt=k_{f}(q_{e}-q_{t}\right)$$
(2)$$\log\left(q_{e}-q_{t}\right)=\log\left(q_{e}\right)-k_{f}t/2.303$$

The pseudo-second-order kinetic model assumes that the adsorption rate is determined by the square value of the number of vacant adsorption spots on the adsorbent surface, and the adsorption process is controlled by the chemisorption mechanism, which involves electron sharing or electron transfer between the adsorbent and the adsorbate [51]. Equation (3) describes this model, in which $q_{e}$ is the equilibrium absorption capacity, $q_{t}$ is the absorption capacity at time *t*, and $k_{s}$ is the constant. Equations (4) and (5) are the integrated forms.
(3)$$(dq_{t})/dt=k_{s}{\left(q_{e}-q_{t}\right)}^{2}$$
(4)$$1/(q_{e}-q_{t})=1/q_{e}+k_{s}t$$
(5)$$t/q_{t}=1/k_{s}{q_{e}}^{2}+t/q_{e}$$

The Weber–Morris dynamics model is a particle diffusion model (Equation (6)) [52,53]. The model is the most suitable for describing the dynamics of the material diffusion process inside the particle, but it is often not suitable for describing the diffusion process outside the particle and inside the liquid film.
(6)$$q_{t}=k_{w}t^{1/2}$$

In Equation (6), $q_{t}$ is the absorption capacity and *k_w_* is the rate constant.

#### 3.5.2. Isotherm Modeling

The adsorption isotherm model is a mathematical model that expresses the relationship between the adsorption capacity and the solution concentration under the condition of fixed temperature. Different types of mathematical expression have been proposed and each has its own scope of application; commonly used expressions are the Langmuir, Freundlich, and Dubinin–Radushkevich isotherm models.

The Langmuir model assumes that the appearance of the adsorbent is uniform and there is no interaction between adsorbents. The adsorption is monolayer adsorption, and the adsorption occurs only on the outer surface of the adsorbent. The Langmuir isothermal adsorption model was the first model to vividly describe the adsorption mechanism [54]. Equations (7) and (8) describe the linearized form [55], where $q_{e}$ is the equilibrium absorption capacity, $C_{e}$ is the equilibrium concentration, $q_{m}$ is the maximum equilibrium absorption capacity, and *b* is the Langmuir rate constant.
(7)$$q_{e}=\frac{q_{m}bC_{e}}{1+bC_{e}}$$
(8)$$\frac{C_{e}}{q_{e}}=\frac{1}{q_{m}b}+\frac{1}{q_{m}}C_{e}$$

Equation (9) is the essential expression for the Langmuir model, and it is easy to estimate the characteristics of adsorption.
(9)$$R_{L}=\frac{1}{1+bC_{0}}$$

$R_{L}$ is a dimensionless constant and $C_{0}$ is the concentration of the Mn(II) solution. $R_{L}$ values between 0 and 1 indicate favorable adsorption [56].

The Freundlich model can be applied to monolayer adsorption and heterogeneous surface adsorption. Not only can the Freundlich adsorption equation describe the adsorption mechanism of an uneven surface, it is also more suitable for adsorption at low concentrations. It can explain the experimental results over a wider range of concentration. The Freundlich isotherm (Equations (10) and (11)) [57] can be expressed as:(10)$$q_{e}=K_{f}{C_{e}}^{1/n}$$
(11)$$lnq_{e}=lnK_{f}+\frac{1}{n}\ln C_{e}$$
where $q_{e}$ is the equilibrium absorption capacity, $C_{e}$ is the equilibrium concentration, $K_{f}$ is the Freundlich constant, and the n value ranges from 1 to 10.

In order to describe the relative pressure–adsorption capacity characteristics of microporous filling, Dubinin and Radushkevich proposed a method, based on the Polanyi adsorption potential theory, to calculate the adsorption characteristics based on the adsorption isotherm in the low-pressure region. The Dubinin–Radushkevich (D-R) isotherm is shown in Equation (12) [58]:(12)$$lnq_{e}=lnq_{m}-K\varepsilon^{2}$$
(13)$$\varepsilon =RTln(1+1/C_{e})$$
where $q_{e}$ is the absorption capacity, $q_{m}$ is the theoretical adsorption capacity, $C_{e}$ is the equilibrium concentration, and K is the D-R constant.

In order to describe the calculation results of this model more easily, the relationship between the free adsorption energy, *E_s_*, and the constant, *K*, is given as Equation (14). The value of *Es* determines the mechanism of isothermal adsorption.
(14)$$E_{s}={\left(2K\right)}^{-0.5}$$

If the value of *E_s_* is between approximately 8 and 16 kJ/mol, it shows that the main process of adsorption is ion exchange, whereas if the value *E_s_* is more than 18 kJ/mol, it means the adsorption is chemisorptive in nature.

#### 3.5.3. PSO-BP Modeling

Since Warren McCulloch and Walter Pitts introduced the concept of artificial neural networks (ANN) in 1943, ANN have evolved rapidly and have been successfully applied in many fields [59]. Backpropagation-based training-optimization neural networks (BPNN) are the most extensively utilized neural networks in practice and are capable of solving complex nonlinear problems. BPNN also have a wide range of applications in the field of pollutant removal [28,29,30]. BPNN require constant exploration of different combinations of weights and biases to obtain ideal results. Particle swarm optimization (PSO) algorithms are widely used for this work. PSO is a stochastic search algorithm that simulates the predatory behavior of birds. A set of optimal solutions was obtained from the PSO algorithm by tracking the constantly moving particles.

In this study, a BPNN model optimized by PSO (PSO–BP) was used to predict the adsorption of Mn onto two carbon materials. This process was implemented with Matlab2016a software. A standard BPNN network with one input layer, one hidden layer, and one output layer was created by calling on the software’s own “newff” function. The backpropagation algorithm, which was applied to determine the most favorable network structure, selected Levenberg–Marquardt (trainlm) with 1000 iterations. The input layer contains four neurons (pH, manganese concentration, time, and temperature). The output is the percentage of Mn(II) removed. Owing to the small amount of data, the network uses just one hidden layer to avoid overfitting [60] and uses a tangent sigmoid function (tansig) as the activation function, whereas the output layer uses a linear function (purlin) as the transfer function. PSO uses the weights and MSE of the network as the particle and fitness functions, respectively, to learn and iterate. After each iteration, the new particles are transformed into the weights of the neural network; the network computes a new MSE and continues to iterate until the result is ideal or the number of iterations reaches a maximum.

The results of the model calculations are evaluated by the following equations (Equations (15)–(18)):(15)$$R^{2}=1-\frac{\sum_{i=1}^{n}{\left(y_{prd,i}-y_{exp,i}\right)}^{2}}{\sum_{i=1}^{n}{\left(y_{prd,i}-y_{m}\right)}^{2}}$$
(16)$$RMSE=\sqrt{\frac{1}{n}\sum_{i=1}^{n}{\left(y_{prd,i}-y_{exp,i}\right)}^{2}}$$
(17)$$MAE=\frac{1}{n}\sum_{i=1}^{n}\left|y_{prd,i}-y_{exp,i}\right|$$
(18)$$MAPE=\frac{100\%}{n}\sum_{i=1}^{n}\left|\frac{y_{prd,i}-y_{exp,i}}{y_{exp,i}}\right|$$
where $y_{prd,i}$ is the value calculated using the PSO–BP model, $y_{exp,i}$ is the experimental value, n is the number of data, and $y_{m}$ is the average of the experimental value.

## 4. Conclusions

Compared with previous studies, the Mn(II)-removal performance by M-MWCNTs was improved [31]. Through the comparative analysis of several analytical techniques results before and after modification, as well as the fitting of thermodynamic and kinetic adsorption models, it was concluded that the increased number of adsorption sites (carboxyl groups and hydroxyl groups) after modification was the key to improving the removal rate. Additionally, the simulation verification of the experimental results by PSO–BP model also provided a scientific guarantee for the reliability of the entire research work.

## Figures and Tables

**Figure 1 molecules-28-01870-f001:**
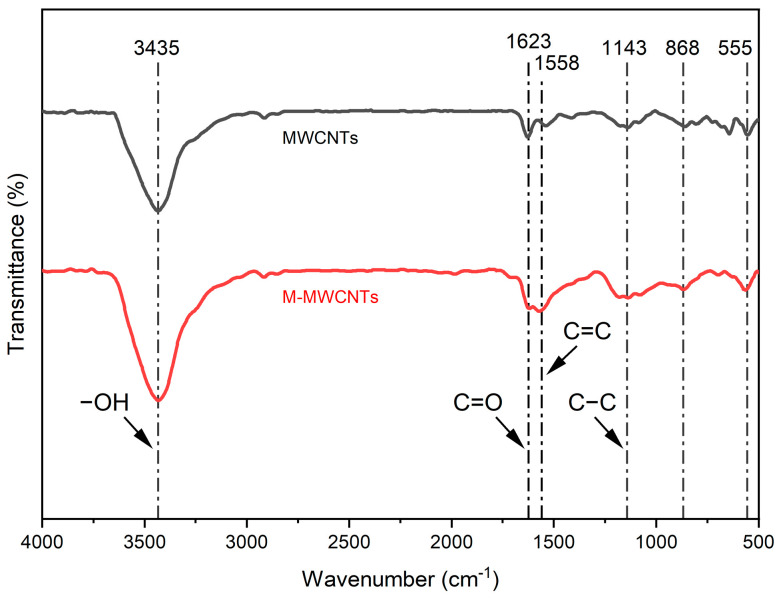
FTIR spectra of MWCNTs and M-MWCNTs.

**Figure 2 molecules-28-01870-f002:**
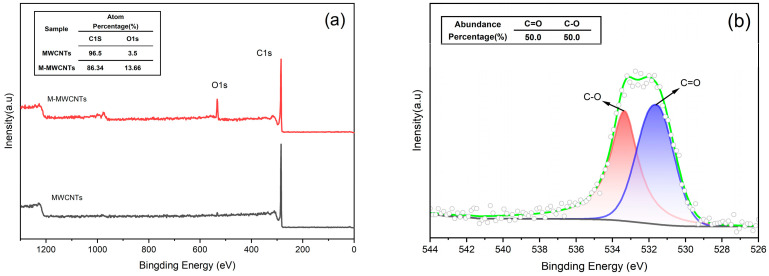
XPS spectra. (**a**) XPS of MWCNTs and M-MWCNTs (**b**) O1s XPS of M-MWCNTs.

**Figure 3 molecules-28-01870-f003:**
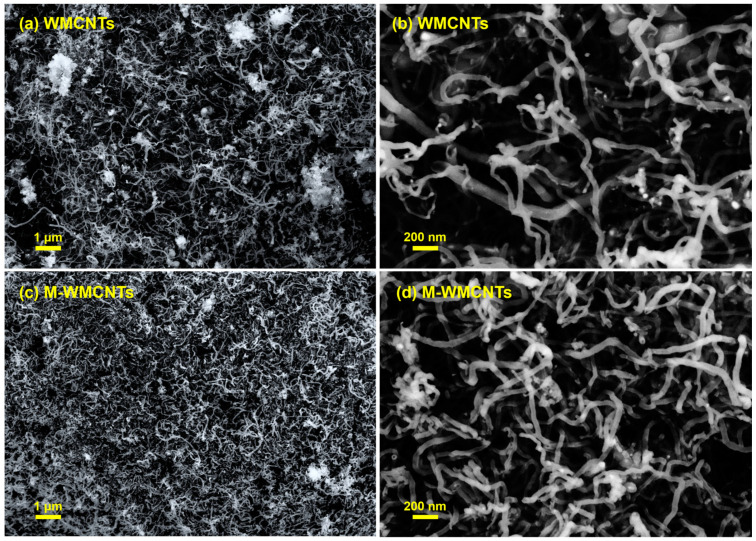
SEM of MWCNTs (**a**,**b**) and M-MWCNTs (**c**,**d**).

**Figure 4 molecules-28-01870-f004:**
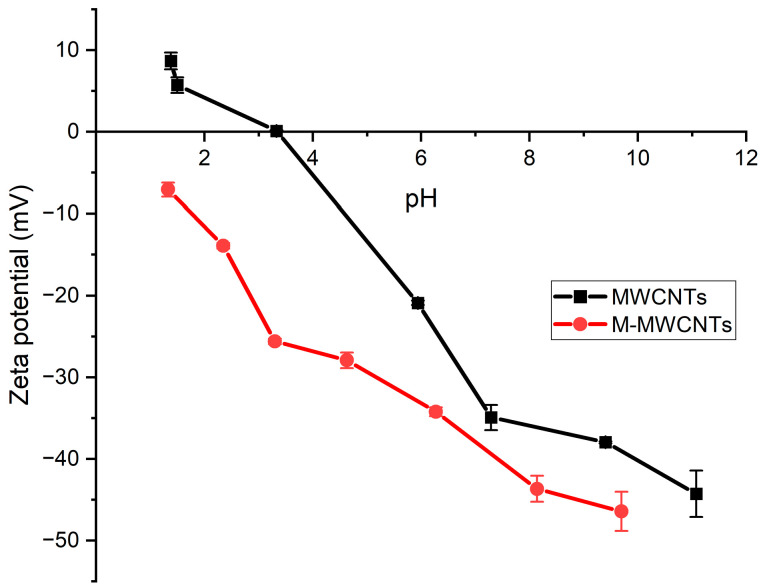
Zeta potentials of MWCNTs and M-MWCNTs.

**Figure 5 molecules-28-01870-f005:**
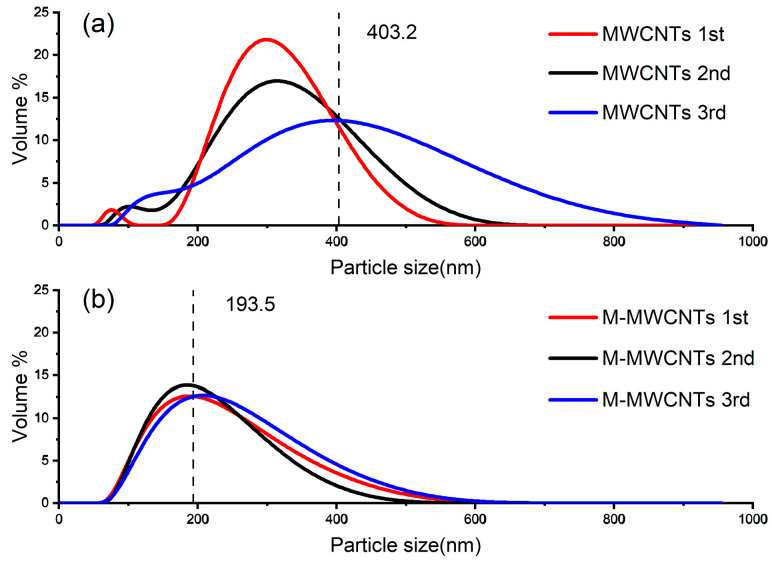
Results of nano-particle sizing (**a**) MWCNTs (**b**) M-MWCNTs.

**Figure 6 molecules-28-01870-f006:**
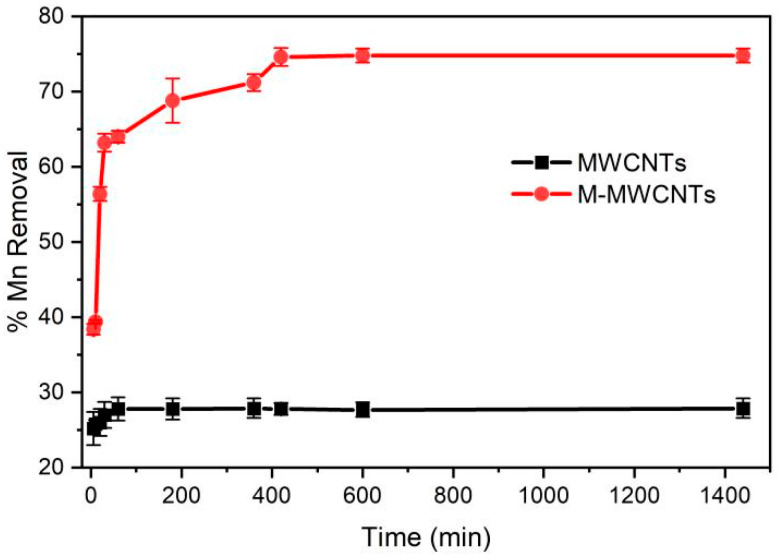
Contact time for Mn(II) adsorption by MWCNTs and M-MWCNTs.

**Figure 7 molecules-28-01870-f007:**
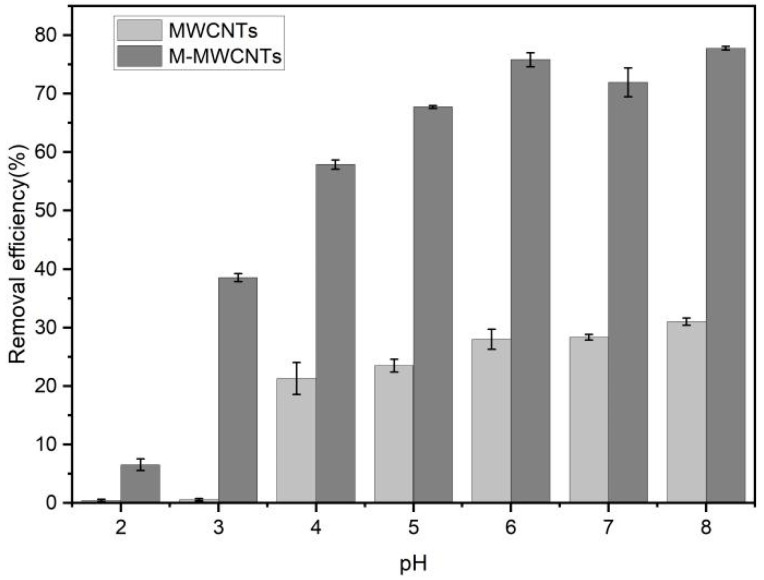
pH and Mn(II) adsorption by MWCNTs and M-MWCNTs.

**Figure 8 molecules-28-01870-f008:**
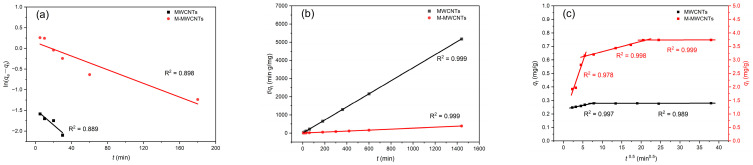
Adsorption kinetics. (**a**) Pseudo-first order, (**b**) pseudo-second order, and (**c**) intraparticle diffusion for Mn(II) adsorption by MWCNTs and M-MWCNTs.

**Figure 9 molecules-28-01870-f009:**
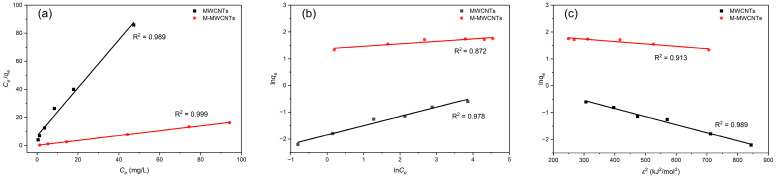
Isotherm models for Mn(II) adsorption by MWCNTs and M-MWCNTs. (**a**) Langmuir, (**b**) Freundlich, and (**c**) D–R.

**Figure 10 molecules-28-01870-f010:**
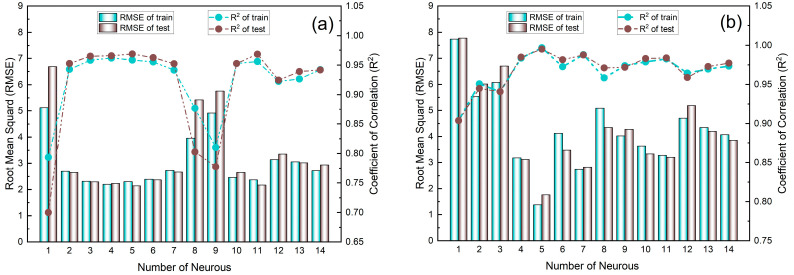
Variation in RMSE and R2 with number of neurons in hidden layer for Model A (**a**) and Model B (**b**).

**Figure 11 molecules-28-01870-f011:**
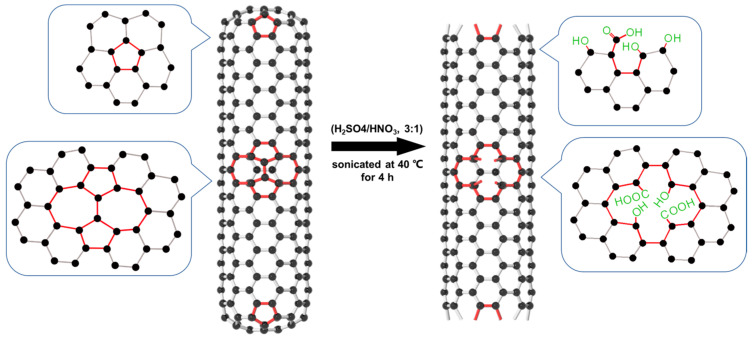
Schematic diagram of the preparation of M-MWCNTs.

**Table 1 molecules-28-01870-t001:** Pseudo-first-order, pseudo-second-order and intra-particle diffusion model constants for Mn(II) adsorption by MWCNTs and M-MWCNTs.

Model	Parameter	MWCNTs	M-MWCNTs
	*q_e.experiment_*	0.278	3.74
Pseudo-first-order model	*q_e.calculate_* (mg/g)	0.229	1.16
	*k_f_* (1/min)	0.0433	0.019
	*R* ^2^	0.889	0.898
Pseudo-second-order model	*q_e.calculate_* (mg/g)	0.279	3.76
	*k_s_* (1/min)	4.12	0.030
	*R* ^2^	0.999	0.999
Weber–Morris	*k_w_*	/	/
	*C*	/	/
	*R* ^2^	/	/

**Table 2 molecules-28-01870-t002:** Isotherm parameters and determination coefficients for Mn(II) adsorption by MWCNTs and M-MWCNTs.

Model	Parameter	MWCNTs	M-MWCNTs
Langmuir isotherm	*q_max_*_1_ (mg/g)	0.585	5.78
	*b* (L/mg)	0.249	1.19
	*R_L_* (L/mg)	0.801	0.456
	*R* ^2^	0.989	0.999
Freundlich isotherm	*K_f_* (mg/g (L/mg)^1/n^)	0.159	3.96
	1/*n*	0.344	0.009
	*R* ^2^	0.978	0.872
Dubinin–Radushkevich	*q_max_*_2_ (mg/g)	1.4083	7.43
	*β*	0.0030	0.0009
	*E_S_*	12.95	23.64
	*R* ^2^	0.989	0.913

**Table 3 molecules-28-01870-t003:** The parameters of the optimum PSO–BP model.

Parameter	Model A: MWCNTs	Model B: M-MWCNTs
Number of samples	117	105
Number of hidden layers	1	1
Hidden nodes	4	5
PSO swarm size	50	50
Cognitive component (c1)	1.5	1.49554
Social component (c2)	1.5	1.49554
Number of iterations	1000	1000

**Table 4 molecules-28-01870-t004:** Nonlinear statistical metrics for validating efficacy of model predictions.

Statistical Metric	Model A: MWCNTs	Model B: M-MWCNTs
*R* ^2^ * (train/test)*	0.962/0.966	0.997/0.995
*RMSE (train/test)*	2.20/2.26	1.38/1.76
*MAE (train/test)*	1.404/1.58	0.769/0.909
*MAPE*	0.074/0.067	0.0202/0.0223

## Data Availability

We can provide the data of this journal through the corresponding author’s email.

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
