# Peer review of "Modification of Multiwalled Carbon Nanotubes and Their Mechanism of Demanganization"

_molecules, 2023, doi:10.3390/molecules28041870_

Round 1
Reviewer 1 Report
In this paper, raw multi-walled carbon nanotubes were modified with concentrated HNO3 and H2SO4, and modified multiwalled carbon nanotubes were characterized by FT-IR, XPS, SEM, EDR and Zeta Potential. It is an interesting content, but arranged structure needs to be further improved. Therefore, it needs minor revision before it is published in this journal. Some issues should be carefully addressed. Please check the attachment.

Author Response
Dear Reviewer,
Thank you for taking time out of your busy schedule to carefully review our article. We have carefully revised your opinions. The details are as follows:
Q1. The English must be improved substantially.
Answer:The grammar in the paper has been modified by a professional organization, and English-Editing-Certificate is on the picture.
Q2. The abstract should be rewritten, i.e., it should highlight the originality, innovation and uniqueness of this paper.
Answer:The abstract has been rewritten. Line 10-23.
Q3. More details about the experimental conditions should be provided.
Answer:The details about the experimental conditions have been supplemented. Line 109-119,317,329
Q4.Analyses of data/results and focus of aims should be further elevated in XPS spectra.
Answer:The contents have been renewed. Line 265-276.
Q5.Put the bands values on the graph of FT-IR.
Answer:The bands value have been added on Figure 2. Line 259
Q6.XPS is an important tool in this work, so the authors should introduce the importance of XPS analysis.
Answer:The contents and references have been supplemented.Line 265-276.464-600(red lines)
Q7.The data in the manuscript should have error analysis.
Answer:The error analysis has been added on the Figure 5,7and 8.Line 296,323 and 343.
Q8. It is suggested to compare the results of the present research with some similar studies which is done before.
Answer:The contents have been supplemented on Conclusions.Line 422-429.
Thank you very much again!
Best regards!
Yan Dou
School of Water and Environment
Chang'an University

Reviewer 2 Report
The study presents the modification of multiwalled CNT and subsequent Demanganization applications. Although interesting, there are several concerns as detailed below:
· The first sentence in the abstract is too long and confusing.
· The authors should also present an overview of why Demanganization is relevant in several industries.
· Several grammatical and spelling ERRORS
· What is the study novelty and how is it different from similar studies? A comprehensive review of past studies and knowledge gaps should be briefly presented in the introduction section
· The following acronyms should either be written in full or removed in the abstract: y FT-IR, XPS, SEM, EDR. The authors could just use “several analytical techniques”
· Why was multiwalled CNT selected?
· The introduction is too short and lacks detailed information on prior study, novelty of the current studies and study objectives
· Procedure for the Manganese stock solution is still not clear. What is the concentration of the stock solution, what mass was dissolved and what is the water volume? What kind of liquid was used?
· The most important concern is that the experiments were performed in a manganese solution, however in real life wastewaters there are other heavy metals whose interactive effect influences the efficiency of Mn removal. How would the authors account for this?
· A schematic diagram is often the best way to explain the Preparation of M-MWCNTs
· It is still not clear how the adsorption experiments were performed.
· Although kinetic modelling is reasonable, the authors should explain the reason for selecting the three models
· The authors claimed some peaks in the FTIR results corresponds to certain functional groups but did not provide literature references/source to validate the claim.
· The XPS confirms that there is no N or S containing functional group. While this is promising, can the results be validated with ICP analysis?
Author Response
Dear Reviewer,
Thank you for taking time out of your busy schedule to carefully review our article. We have carefully revised our paper. The details are as follows:
Q1.The first sentence in the abstract is too long and confusing.
Answer: The first sentence in the abstract has been modified.Line 10-12.
Q2.The authors should also present an overview of why Demanganization is relevant in several industries.
Answer:The contents have been supplemented on Line 27-37.
Q3.Several grammatical and spelling ERRORS
Answer:The grammar in the paper has been modified by a professional organization, and English-Editing-Certificate is on the picture.
Q4.What is the study novelty and how is it different from similar studies? A comprehensive review of past studies and knowledge gaps should be briefly presented in the introduction section
Answer:The contents have been supplemented on Line 56-58.
Q5.The following acronyms should either be written in full or removed in the abstract: y FT-IR, XPS, SEM, EDR. The authors could just use “several analytical techniques”
Answer:The sentence has been supplemented on Line 12.
Q6.Why was multiwalled CNT selected?
Answer:The contents have been supplemented on Line 45-47.
Q7.The introduction is too short and lacks detailed information on prior study, novelty of the current studies and study objectives
Answer:The contents have been supplemented on Line 47-55.
Q8.Procedure for the Manganese stock solution is still not clear. What is the concentration of the stock solution, what mass was dissolved and what is the water volume? What kind of liquid was used?
Answer:The contents have been renewed. Line 265-276.
Q9.The most important concern is that the experiments were performed in a manganese solution, however in real life wastewaters there are other heavy metals whose interactive effect influences the efficiency of Mn removal. How would the authors account for this?
Answer:This research is mainly focused on the manganese removal performance of new material--modified multi-walled carbon nanotubes. However, the coexistence of manganese, iron, arsenic and other ions has been considered in our work, and this part of the research is still continuing.
Q10.A schematic diagram is often the best way to explain the Preparation of M-MWCNTs
Answer:Figure 1 Schematic diagram of the preparation of M-MWCNTs has been added.
Q11.It is still not clear how the adsorption experiments were performed.
Answer:The contents have been renewed. Line 109-118.
Q12.Although kinetic modelling is reasonable, the authors should explain the reason for selecting the three models
Answer:The contents have been renewed. Line 124-132.
Q13.The authors claimed some peaks in the FTIR results corresponds to certain functional groups but did not provide literature references/source to validate the claim.
Answer:The contents have been renewed. Line 248-257.
Q14.The XPS confirms that there is no N or S containing functional group. While this is promising, can the results be validated with ICP analysis?
Answer: On the manuscript, we wrote the sentence “The oxidation of carbon nanotubes using mixed acids did not introduce N or S-containing functional groups.”, we totally agree reviewer’s opinion, our description is not precise enough, so we have deleted this sentence in this article. In the future research work, we will study this problem in more detail.
Thank you very much again!
Best regards!
Yan Dou
School of Water and Environment
Chang'an University

Reviewer 3 Report
Report on the manuscript molecules-2189876 entitled “Modification of Multiwalled Carbon Nanotubes and its Mechanism in Demanganization”.
The submitted manuscript should be revised. The following points should be addressed:
1. The submitted manuscript should be revised to be free from editing or grammar errors.
2. The experimental part should have more details such as why the authors use pH 6.
3. In FT-IR analysis, the C=O bond has a low wavenumber around 1600 and it’s reported to be around 1700 so, the authors could indicate why it’s appeared as lower than expected.
4. The XPS peaks data should supported by references [suggested one: Applied Surface Science
Volume 400, 2017, Pages 355-364 & Composites Science and Technology, Volume 208, 2021, 108753 & Ceramics International, 46 (3), 2020, 3912-3920]
5. The SEM analysis should have a high magnification image beside low-one to confirm the morphology.
6. The conclusion should have the main achievement of the study.
Author Response
Dear Reviewer,
Thank you for taking time out of your busy schedule to carefully review our article. We have carefully revised your opinions. The details are as follows:
1.The submitted manuscript should be revised to be free from editing or grammar errors.
Answer:The grammar in the paper has been modified by a professional organization, and English-Editing-Certificate is on the picture.
- The experimental part should have more details such as why the authors use pH 6.
Answer:Some details of experiments had been added. Line 113
- In FT-IR analysis, the C=O bond has a low wavenumber around 1600 and it’s reported to be around 1700 so, the authors could indicate why it’s appeared as lower than expected.
Answer:The contents have been supplemented on Line 248-258.
- The XPS peaks data should supported by references.
Answer:The References have been supplemented.
- The SEM analysis should have a high magnification image beside low-one to confirm the morphology.
Answer:The contents have been supplemented on Figure 4, Line 277.
- The conclusion should have the main achievement of the study.
Answer:The conclusion has been rewrote on Line 420-427.
Thank you very much again!
Best regards!
Yan Dou
School of Water and Environment
Chang'an University

Round 2
Reviewer 2 Report
The authors have addressed all concerns
Reviewer 3 Report
The revised version of the manuscript could be accepted.